# Smoothed Maximum Score Estimation of Discrete Duration Models

**Sadat Reza [1] and Paul Rilstone [2],***

[1]  Nanyang Business School, Nanyang Technological University, Singapore 639798, Singapore; SReza@ntu.edu.sg
[2]  Department of Economics, York University, Toronto, ON M3J 1P3, Canada
*  Correspondence: pril@yorku.ca

**Abstract:** This paper extends Horowitz's smoothed maximum score estimator to discrete-time duration models. The estimator's consistency and asymptotic distribution are derived. Monte Carlo simulations using various data generating processes with varying error distributions and shapes of the hazard rate are conducted to examine the finite sample properties of the estimator. The bias-corrected estimator performs reasonably well for the models considered with moderately-sized samples.

**Keywords:** maximum score estimator; discrete duration models; efficient semiparamteric estimation

---

## 1. Introduction

Parametric discrete-time duration models are used extensively within econometrics and the other statistical sciences. Since misspecification of these models can lead to invalid inferences, a variety of semiparametric alternatives have been proposed. However, even these alternative semiparametric estimators exploit certain smoothness and moment conditions, which may be untenable in some circumstances. To address these shortcomings, we propose a new estimator, based on Horowitz (1992)'s smoothed maximum score estimator of single-period binary choice models, which relaxes these assumptions. To motivate and contextualize this estimator, we use this Introduction to review the relevant literature on discrete duration and binary choice models and indicate how our proposed estimator fills a gap in the literature.

In econometrics, discrete-time duration models are typically framed as a sequence of binary choices. The probability of remaining in a state at time $s$ (the continuation probability) is denoted $F_s(\beta_0)$, and the hazard rate is simply $h_s(\beta_0) = 1 - F_s(\beta_0)$. Many parametric forms have been employed for the hazard rate in these models including extreme value, logistic, normal and other parsimonious specifications. Examples using a logistic specification include: Huff-Stevens (1999), Finnie and Gray (2002), Bover et al. (2002) and D'Addio and Rosholm (2005); normal distribution: Meghir and Whitehouse (1997) and Chan and Huff-Stevens (2001); extreme value (also known as the complementary log-log model): Baker and Rea (1998), Cooper et al. (1999), Holmas (2002), Fennema et al. (2006) and Gullstrand and Tezic (2008). These and others were reviewed in Allison (1982) and Sueyoshi (1995). Hess (2009) has suggested using the generalized Pareto distribution, which nests the extreme value and logistic distributions. These specifications lead naturally to maximum likelihood estimation of $\beta_0$, although it is useful to note that there are alternative ways to estimate $\beta_0$ including nonlinear regression, treating $F_s(\beta_0)$ as a conditional mean. As with any parametric approach, misspecification of the hazard rate can lead to invalid inferences. In this regard, we consider various relevant semiparametric alternatives, which relax the parametric assumptions.

We note first that semiparametric estimation of continuous-time models has been the focus of substantial research in the discipline. Numerous authors have developed distribution theory for

semiparametric estimation of various continuous-time duration models including Horowitz (1999), Nielsen et al. (1998), Van der Vaart (1996) and Bearse et al. (2007). While these and other semiparametric estimators allow for the relaxation of some parametric assumptions associated with continuous-time duration models, they are not generally appropriate when the duration random variable has a discrete distribution.

We adopt the standard approach in econometrics of constructing the continuation probability from an underlying latent regression structure. In a standard single-period basic binary choice model, we would observe $Y = 1[Y^* \geq 0]$ with $Y^* = Z + U$ where $1[\cdot]$ is the usual indicator function, $Z$ is an index function of observable random variables and unknown parameters and $U$ has a distribution function $F$. With discrete-time duration models, the observed duration is the sum of a sequence of indicators so that $T = \sum_{s=1}^{S} Y_s$, where $Y_s = Y_{s-1}1[Z_s + U_s > 0]$ with $Y_0 = 1$, and the distribution function of $U_s$ is denoted by $F_s$.

There is a large literature on semiparametric estimation of single-period binary choice models. We briefly review this, highlighting how it has been adapted for certain multivariate discrete choice and/or discrete-duration models and finally how our proposed estimator fills a gap in this research. Since in some cases, the conditional mean of $Y$ in the single-period case can be written as $F(\beta_0)$, the parameter of interest, $\beta_0$, can be estimated from a semi-parametric regression. This was suggested by Ichimura (1993) to obtain a $\sqrt{N}$-consistent estimator of $\beta_0$. With respect to duration models and exploiting the fact that $F_s$ can also be written as the conditional mean of the choice variable, Reza and Rilstone (2014) minimized a sum of squared semiparametric residuals to estimate the parameters of interest. In a similar vein, Klein and Spady (1993) developed a semi-parametric maximum likelihood estimator of $\beta_0$ with the single observation likelihood function written as $l(\beta) = F(\beta)^Y (1 - F(\beta))^{1-Y}$. Klein and Spady's (1993) estimator essentially consists of replacing $F$ with a nonparametric conditional mean function. Reza and Rilstone (2016) adapted Klein and Spady's (1993) estimator to the discrete duration case. They also derived the efficiency bounds and showed that their estimator obtained these bounds. We note that the approaches in Ichimura (1993) and Klein and Spady (1993) require continuity of $F$ in the underlying covariates and are limited with respect to the forms of allowable heteroskedasticity (for example, heteroskedasticity from time-varying parameters is precluded). Another problem is simply that identification may not be possible under the mean-independence restriction that $\mathbb{E}[U|Z] = 0$.[1] By extension, the estimators of Reza and Rilstone (2014, 2016) suffer the same shortcomings as applied to duration models.

With respect to single-period binary choice models, Manski's (1975, 1985) Maximum Score (MS) estimator circumvents these limitations using simply the median-independence restriction that $\text{Median}[U|Z] = 0$. The MS estimator can be written as the maximizer of:

$$\Psi_N^*(\beta) = \frac{1}{N} \sum_{i=1}^{N} (2Y_i - 1)1[Z_i(\beta) > 0] \tag{1}$$

where $Z_i(\beta)$ is an index function of the observable covariates. As is usually the case, a normalization of $\beta$ is necessary. For the estimator to be consistent, a few restrictions need to be imposed, in particular with respect to the distribution of $U$. The shortcomings of the estimator are that it is only $N^{1/3}$-consistent, and its asymptotic distribution, a form of Brownian motion, is not amenable for use in the applied work.

From one perspective, the shortcomings of the MS estimator derive from its use of the non-differentiable indicator function. Horowitz (1992) largely circumvented its limitations in this regard by replacing the indicator function with a smoothed indicator function, $K^{\dagger}(Z_i(\beta)/\gamma)$. The objective function for the Smoothed Maximum Score (SMS) estimator is:

---

[1]    Horowitz (1998) gave a discussion of these issues.

$$\Psi_N(\beta) = \frac{1}{N} \sum_{i=1}^{N} (2Y_i - 1) K^{\dagger}(Z_i(\beta)/\gamma). \tag{2}$$

The SMS is typically better than $N^{1/3}$-consistent, but slower than $\sqrt{N}$, the speed of convergence depending on the smoothness of $K^{\dagger}$ and the distribution of the random components of the model. Note that the $\sqrt{N}$-convergence of the estimators such as Klein and Spady's (1993) is linked to the manner in which they use kernels. These estimators are a form of double averages. However, the objective functions for MS and SMS are nonparametric point estimators, which are single averages. With some caveats, the SMS estimator reflects the fact that the only exploitable information is at or close to the median of the $U$'s. The $\sqrt{N}$ estimators effectively use all the data points.

The main objective of this paper is to show how to extend SMS to estimate discrete duration models. The MS and SMS estimators have been used in other situations such as Lee (1992) and Melenberg and Van Soest (1996), who extended the MS and SMS, respectively, to ordered-response models. De Jong and Woutersen (2011) have extended the SMS estimator to binary choices with dynamic time series data. Fox (2007) adapted the MS estimator to multinomial choices. Charlier et al. (1995) extended the SMS to panel data. Other researchers have modified the MS and SMS estimators to improve their sampling properties. Kotlyarova and Zinde-Walsh (2010) suggested using a weighted average of different SMS estimators to reduce mean squared error. Iglesias (2010) derived the second-order bias, which can be used to reduce the bias of the SMS estimator. Jun et al. (2015) proposed a Laplace estimator alternative to improve on the $N^{1/3}$-consistency of the MS estimator. To our knowledge, neither the MS nor SMS estimators have been extended to duration models.

Sections 2 and 3 discuss the class of models considered and present the basic estimator along with its main asymptotic properties. Section 4 provides some simulation results concerning the sampling distribution of the estimator, and Section 5 concludes.

## 2. Modelling

As mentioned, a standard approach for modelling a discrete duration process is to construct it as a sequence of binary choice models, with observed and unobserved heterogeneity. The standard binary choice model is adapted such that in each time period, $s$, a choice is made by individual $i$ to continue in a state if the latent variable:

$$Y_{is}^* = Z_{is}(\beta_0) + U_{is}, \qquad s = 1, 2, \ldots, S \tag{3}$$

is greater than zero. Here, $Z_{is}(\beta) = X_{is}^* + X_{is}^{\top} \beta^2$ is an index where $X_{is}^*$ is a scalar random variable and $X_{is}$ is a $k \times 1$ vector, which may include a function of $s$, while $\beta$ is a $k \times 1$ vector of constants.

We assume the $U_{is}$'s and $X_{is}^*, X_{is}$'s are jointly i.i.d. We observe $Y_{is} = 1[Y_{is}^* > 0]Y_{is-1}$ and $X_{is}^*, X_{is}, s = 1, \ldots, S$. A natural adaptation of Manski's setup is the additional assumption that $\text{Median}[U_s | X_s, Y_{s-1}] = 0, s = 1, \ldots, S$. We estimate the parameters by effectively estimating the density of $Z_{is}(\beta_0)$ at zero by nonparametric methods. For notational convenience, we often suppress the $i$ subscripts. Another way to view the modelling is that in any given period $s$ with $Y_{s-1} = 1$, this is a standard binary choice variable with the key difference being that the index $Z$ is a function of some covariates and the number of completed periods, $s$. The duration variable for period $s$ is simply $T_s = \sum_{j=0}^{s-1} Y_j$ with $Y_0 = 1$, $Y_{S+1} = 0$.[3] The evolution of the $Y_s$'s, conditional on the covariates and duration, is given by:

$$Y_s = 1[Z_s(\beta_0) + U_s \geq 0]Y_{s-1}, \qquad s = 1, \ldots, S. \tag{4}$$

---

[2]　Some normalization of the parameter space is necessary. We find it most convenient to impose a unit coefficient on $X_{is}^*$ immediately.

[3]　The model is easily reformulated to incorporate functions of the $Y_j$'s, $j \leq s$ as conditioning variables.

Note that this representation is such that $Y_s$ is zero if the subject left the state prior to period $s$ and becomes a standard binary choice model in period $s$ if the subject elected to continue in the state in period $s - 1$.

We put an upper limit, $S$, on the length of spells. This is common in empirical work.[4] Allowing for unbounded $S$ introduces technical difficulties that are not readily resolved. Put $\mathcal{Z}_s = \{X_{ij}^*, X_{ij}, Y_{i,j-1}\}_{j=1}^s$. It is useful to note that by iterated expectations:

$$\mathbb{E}[Y_s|\mathcal{Z}_s] = \mathbb{E}[Y_s|Z_s(\beta_0), Y_{s-1}] = F_s Y_{s-1} \tag{5}$$

so that, tautologically, $F_s$, the continuation probability function, is:

$$F_s = \mathbb{E}[Y_s|Z_s(\beta_0), Y_{s-1} = 1] = \Pr[Y_s = 1|Z_s(\beta_0), Y_{s-1} = 1]. \tag{6}$$

## 3. The Estimator

Adapting the SMS estimator to the discrete duration model as outlined in Section 2, the objective function is:

$$\Psi_N(\beta) = \frac{1}{N} \sum_{i=1}^N \sum_{s=1}^S Y_{is-1}(2Y_{is} - 1)K^\dagger(Z_{is}(\beta)/\gamma). \tag{7}$$

$K^\dagger(w)$, a smoothed indicator function, is the anti-derivative of $K(w) = dK^\dagger(w)/dw$ and has the properties: $|K^\dagger(w)| \le M < \infty$, $\lim_{w \to -\infty} K^\dagger(w) = 0$, $\lim_{w \to \infty} K^\dagger(w) = 1$. In most kernel density estimation, $K$ is a density function and $K^\dagger$ is its associated cumulative distribution function. The technical requirements here sometimes require use of a higher order kernel.

Note that the objective function is of the same form as the usual SMS estimator with the modifications that there is a double summand over individuals and time periods and each of the summands at period $s$ is multiplied by $Y_{s-1}$, so that after exit, there is no further contribution to the objective function by that individual.

Implicitly, we impose the identification condition that the coefficient on $X_{is}^*$ is unity[5] (e.g., Li and Racine 2007). Horowitz (1992) discussed the identification issue. $X_{is}^*$ is assumed to have a continuous distribution, conditional on $X_{is}$ and $Y_{is-1}$. Let:

$$Y_i = \begin{pmatrix} Y_{i1} \\ \vdots \\ Y_{iS} \end{pmatrix}, \quad X_i = \begin{pmatrix} X_{i1} \\ \vdots \\ X_{iS} \end{pmatrix}, \quad X_i^* = \begin{pmatrix} X_{i1}^* \\ \vdots \\ X_{iS}^* \end{pmatrix}, \quad Z_i = \begin{pmatrix} Z_{i1} \\ \vdots \\ Z_{iS} \end{pmatrix}. \tag{8}$$

The estimator solves the first-order conditions $\psi_N(\widehat{\beta}) = 0$, which are given by:

$$\psi_N(\beta) = \frac{1}{N} \sum_{i=1}^N q_i(\beta), \quad q_i(\beta) = \sum_{s=1}^S q_{is}(\beta),$$

$$q_{is}(\beta) = Y_{is-1}(2Y_{is} - 1)\frac{1}{\gamma}K\left(\frac{Z_{is}(\beta)}{\gamma}\right)X_{is}. \tag{9}$$

Concerning notation, when a function's argument $\beta$ is suppressed, it is evaluated at $\beta_0$, e.g., $q_i = q_i(\beta_0)$. $q_i^{(1)}(\beta) = \partial q_i(\beta)/\partial \beta^\top$, a $k \times k$ matrix. Thus,

---

[4] For example, Cameron and Heckman (1998) defined $S$ as the upper limit to years of education. In practice, for programming purposes, it suffices to set $S$ equal to the longest duration in the dataset being used. In the simulations reported in Section 4, the maximum duration was 37.

[5] This has two aspects: one is that it implies that estimates of the other $\beta$'s are all to scale and that we know the sign of the first coefficient.

$$\psi_N^{(1)}(\beta) = \frac{1}{N}\sum_{i=1}^{N} q_i^{(1)}(\beta), \quad q_i^{(1)}(\beta) = \sum_{s=1}^{S} q_{is}^{(1)}(\beta),$$

$$q_{is}^{(1)}(\beta) = Y_{is-1}(2Y_{is}-1)\frac{1}{\gamma^2}K^{(1)}\left(\frac{Z_{is}(\beta)}{\gamma}\right)X_{is}X_{is}^{\top}.$$

$G(u_s|z_s, x_s, y_{s-1})$ and $g(u_s|z_s, x_s, y_{s-1})$ denote the cumulative distribution and density functions of $U_s$ conditional on $Z_s, X_s, Y_{s-1} = 1$, and $f(z_s|x_s, y_{s-1})$ denotes the density functions of $Z_s$ conditional on $X_s, Y_{s-1}$. The superscript $[j]$ indicates the $j^{\text{th}}$ derivative of a function with respect to $z_s$, and in particular, we have $G^{[j]}(-z_s|z_s, x_s, y_{s-1}) = d^j G(-z_s|z_s, x_s, y_{s-1})/dz_s^j$. $0 \le M < \infty$ is a generic constant. Put:

$$B = -2\frac{\mu_m}{m!}\mathbb{E}\left[\sum_{s=1}^{S}\sum_{j=1}^{m}\binom{m}{j}G^{[j]}(0|0, X_s, Y_{s-1})f^{[m-j]}(0|X_s, Y_{s-1})X_s Y_{s-1}\right],$$

$$C = \mathbb{E}\left[\sum_{s=1}^{S} f(0|X_s, Y_{s-1})X_s X_s^{\top}Y_{s-1}\right]\int K(w)^2 dw, \tag{10}$$

$$Q = 2\mathbb{E}\left[\sum_{s=1}^{S} G^{[1]}(0|0, X_s, Y_{s-1})f(0|X_s Y_{s-1})X_s X_s^{\top}Y_{s-1}\right].$$

Let $\Pr[u_s, x_s, x_s^*|\mathcal{Z}_{-1}]$ denote the probability distribution of $U_{is}, X_{is}, X_{is}^*$ given $\mathcal{Z}_{i,s-1}$. The distributional assumptions we make are as follows.

**Assumption 1.** $\{Y_i, X_i, X_i^*\}_{i=1}^{N}$ *is a random sample where* $Y_{is} = 1[Z_{is}(\beta_0) + U_{is} \ge 0]Y_{is-1}$. $\Pr[u_s, x_s, x_s^*|\mathcal{Z}_{-1}] = \Pr[u_s, x_s, x_s^*|Y_{s-1}]$. $Z_{is}(\beta) = X_{is}^* + X_{is}^{\top}\beta$. $Y_{i0} = 1$ *for all* $i$.

**Assumption 2.** *For* $s = 1, \ldots, S$, *(a) the support of the distribution of* $x_s^*, x_s$ *is not contained in any proper linear subspace of* $\mathbb{R}^{k+1}$, *(b)* $0 < \Pr(y_s = 1|x_s^*, x_s, y_{s-1} = 1) < 1$ *for almost every* $x_s^*, x_s$ *and (c) for almost every* $x_s, y_{s-1}$, *the distribution of* $x_s^*$ *conditional on* $x_s, y_{s-1}$ *has everywhere positive density with respect to the Lebesgue measure.*

**Assumption 3.** *Median* $(u_s|x_s^*, x_s, Y_{s-1}) = 0$ *for almost every* $x_s, Y_{s-1}, s = 1, \ldots, S$.

**Assumption 4.** $\beta_0 \in \mathcal{B}$, *a compact subset of* $\mathbb{R}^k$.

**Assumption 5.** *The elements of* $X_s$ *have finite fourth moments,* $s = 1, \ldots, S$ .

**Assumption 6.** $(\log N)/(N\gamma^4) \to 0$ *as* $N \to \infty$

**Assumption 7.** *(a)* $K^{\dagger}$ *is twice differentiable everywhere;* $K$ *and* $K^{[1]}$ *are uniformly bounded; and each of the following integrals over* $(-\infty, \infty)$ *is finite:* $\int K(w)^4 dw$ , $\int [K^{[1]}(w)]^2 dw$, $\int lw^2 K^{[1]}(w)|dw$. *(b) For some integer* $m > 2$ *and each integer* $j$, $j = 2, \ldots, m-1$ $\int w^j K(w)dw = 0$, $\int w^m K(w)dw = \mu_m$, $|\mu_m| < \infty$. *(c) For* $j = 2, \ldots, m-1, \gamma \to 0$, *any* $\eta > 0$, $\gamma^{j-m}\int_{|\gamma w|>\eta}|w^j K(w)|dw \to 0$, $\gamma^{-1}\int_{|\gamma w|>\eta}|K^{[1]}(w)|dw \to 0$

**Assumption 8.** $f(z_s|x_s, y_{s-1})$ *is* $m$-*times continuously differentiable with respect to* $z$ *in a neighbourhood of zero, almost every* $x_s, y_{s-1}$, *and* $|f^{[j]}(-z_s|z_s, x_s, y_{s-1})| \le M, s = 1, \ldots, S$.

**Assumption 9.** $G(-z_s|z_s, x_s, y_{s-1})$ *is* $m$-*times continuously differentiable with respect to* $z_s$ *in a neighbourhood of zero, almost every* $x_s, y_{s-1}$ *and* $|G^{[j]}(-z_s|z_s, x_s, y_{s-1})| < M, j = 1, \ldots, m, s = 1, \ldots, S$.

**Assumption 10.** $\beta_0$ *is an interior point of* $\mathcal{B}$.

**Assumption 11.** $Q$ *is negative definite.*

These assumptions adapt those in Horowitz (1992) to allow for the dependency structure. They also embed Manski's (1985) assumptions with $S = 1$. Notice that the random sampling assumption refers to $N$ random draws within each being the potentially $S$ observations.

Identification (see Proof of Proposition 1 in Appendix A) follows by adapting Manski's (1985) proof for the MS estimator. Of interest here is that we wish to allow for time dependence. Note that for the MS/SMS case, nothing precludes the inclusion of a constant in the index so long as, say, $x_s$ is not co-linear[6] (in fact, simulation and empirical results such as in Horowitz (1998) indicate good results for intercept estimates). For the $m$-multinomial choice model, Lee (1992) included $m$ non-stochastic threshold parameters (including a constant). In our case, the same applies for including certain non-stochastic functions of $s$ in $x_s$, such as including indicators for each $s$ or a polynomial in $s$. For parsimony in our numerical/empirical work, we have included quadratics to allow for increasing, decreasing and non-monotonic time dependency. This allows for straight-forward testing. In this regard, we note that the semiparametric information matrix derived in Reza and Rilstone (2016) was singular for this class of models. There is no contradiction here, since the singularity indicates that those parameters are not estimable at the $\sqrt{N}$-rate; it does not imply that they cannot be identified or estimated at a less than $\sqrt{N}$-rate, which we do here.

We have the following lemma, which permits simple derivation of the asymptotic properties of the estimator.

**Lemma 1.** *Let Assumptions 1–11 hold. Then, (a)* $\mathbb{E}[q_i^{(1)}(\beta_0)] = Q + o(1)$, *(b)* $\gamma^m \mathbb{E}[q_i(\beta_0)] = B + o(1)$ *and (c)* $\gamma \mathbb{E}[q_i(\beta_0)q_i(\beta_0)^\top] = C + o(1)$.

The asymptotic distribution of the estimator can be summarized easily using the following result.

**Proposition 1.** *Let Assumptions 1–11 hold. Then, (a)* $\widehat{\beta}$ *is consistent and (b)* $\sqrt{N\gamma}(\widehat{\beta} - \beta_0 - \gamma^m Q^{-1} B) \xrightarrow{d} N(0, Q^{-1} C Q^{-1})$.

The proofs are in Appendix A. In the statement of the proposition, note the presence of the first-order bias, $\gamma^m Q^{-1} B$, for which it may be advisable to adjust the raw estimator. One of the benefits of this estimator is that one can effectively ignore the dependence of the observations, pool all the observations across individuals for whose $Y_{i,s-1} = 1$ and use standard SMS optimization procedures. This is what we have done in the simulations. Reza and Rilstone's (2016) setup (extension of Klein and Spady 1993) allows for estimation of the hazard rate, $1 - F_s$, with a natural estimate of time dependence from the semiparametric estimates of $\Delta h_s = F_{s-1} - F_s$. Note that Reza and Rilstone's (2016) estimator of $\Delta h_s$ only has a $\sqrt{N\gamma}$-rate of convergence.

As for the SMS estimator, we can consider the optimal choice of window width. As with Horowitz (1992), we consider choices that minimize an MSE criterion. Therefore, if we consider that the asymptotic results correspond to the distribution of a random variable, say $W$, with mean $\gamma^m Q^{-1} B$ and variance $Q^{-1} C Q^{-1}/(N\gamma)$, we can consider minimizing, say, the inner product MSE of $\Omega^{1/2} W$, where $\Omega$ is a positive definite weighting matrix, i.e., minimize $\mathbb{E}[W^\top \Omega W]$ with respect to $\gamma$. This results in:

$$\gamma^* = \arg \min MSE(\gamma), \quad MSE(\gamma) = \gamma^{2m} B^\top Q^{-1\top} \Omega Q^{-1} B + \frac{1}{N\gamma} \text{Trace} \left[ \Omega Q^{-1} C Q^{-1} \right] \quad (11)$$

$$\gamma^* = N^{-1/(2m+1)} \left( \frac{\text{Trace}[\Omega Q^{-1} C Q^{-1}]}{2m B^\top Q^{-1\top} \Omega Q^{-1} B} \right)^{1/(2m+1)}. \quad (12)$$

---

6   In this case, the random sampling assumption should be interpreted as referring to the stochastic elements of $x_s$.

For inferences it is necessary to obtain consistent estimates of the components of the first-order bias and variance. These cannot be directly estimated as they depend on the distribution of the unobservable $U$'s. However, by extension of the arguments in Horowitz (1992), they may be obtained through various derivatives of the objective function. Specifically, put:

$$\widehat{B}(\widehat{\beta}) = \frac{1}{\gamma^m} \psi_N(\widehat{\beta}), \quad \widehat{Q}(\widehat{\beta}) = \psi_N^{(1)}(\widehat{\beta})$$

$$\widehat{C}(\widehat{\beta}) = \frac{1}{N\gamma} \sum_{i=1}^{N} \sum_{s=1}^{S} q_{is}(\widehat{\beta}) X_{is}^{\top} K(Z_{is}(\widehat{\beta})/\gamma). \tag{13}$$

By the uniform law of large numbers, $\widehat{B}(\widehat{\beta}) \overset{p}{\to} B$, $\widehat{Q}(\widehat{\beta}) \overset{p}{\to} Q$ and $\widehat{C}(\widehat{\beta}) \overset{p}{\to} C$.

It is well known that the first-order asymptotic results may provide a poor approximation to the sampling distribution of the SMS estimator. Thus, it may be preferable to use some higher order method to approximate the distribution. Apart from Iglesias (2010) who applied the results in Rilstone et al. (1996) to derive the second-order bias of $\widehat{\beta}$, little is known (explicitly) about the second-order properties of the SMS estimator. Estimates can be bootstrapped. In this regard, we note that one should resample individuals. That is, bootstrap estimates should be based on resamples: $\{\mathcal{Z}_{iS}^*\}_{i=1}^{N}$, where the $*$'s indicate random draws from the original data. Horowitz (2002) documents some of the issues associated with bootstrapping the distribution of $\widehat{\beta}$. In particular, the corresponding re-estimates: $\widehat{\beta}_j^*$, say, and corresponding standard errors should be calculated using an under-smoothing window-width such as $\gamma \in [.5\gamma^*, \gamma^*]$.

## 4. Simulation Exercise

To examine the estimator's performance in finite samples, we conducted Monte Carlo simulations with several Data Generating Processes (DGPs). We adapted simulations in Horowitz (1992) by augmenting the models with duration dependence, and a variety of error distributions. The latent processes we considered included those with homoskedastic errors:

$$Y_{is}^* = 1.5 + 2(s/100) - (s/100)^2 + X_{1is} + X_{2is} - u_{is}, \tag{14}$$
$$u_{is} \sim N(0,1)$$

and those with heteroskedastic errors:

$$Y_{is}^* = 1.5 + 2(s/100) - (s/100)^2 + X_{1is} + X_{2is} - v_{is},$$
$$v_{is} = 0.25(1 + (X_{1is} + X_{2is})^2) \cdot u_{is}, \tag{15}$$
$$u_{is} \sim N(0,1).$$

We conducted the simulations for two sample sizes, $N = 500$ and $N = 1000$. The $X$'s were drawn as i.i.d. $N(0,1)$. For the DGP with homoskedastic normal errors, this resulted in duration times with averages of 5.7 ($N = 500, 1000$) and standard deviations also 5.7 ($N = 500, 1000$). With heteroskedastic errors, the average duration times were 8.7 ($N = 500, 1000$) with standard deviations of 9.6 ($N = 500$) and 9.5 ($N = 1000$). For identification purposes, the coefficient on $X_1$ was normalized to one, and our key parameter of interest was the coefficient on $X_2$, with a true value of one. We conducted 500 replications for each specification. We followed Horowitz (1992) to estimate the parameters in two steps: first using simulated annealing to find the approximate maximizer of $\Psi_N(\beta)$ followed by gradient methods for greater precision. We then used the bias correction described in the previous section to bias-adjust the parameter estimates. We used a Gaussian kernel with a window-width

$\gamma = N^{-1/6}$.[7] Standard errors and the bias correction were based on the consistent estimators $\widehat{B}(\widehat{\beta})$, $\widehat{Q}(\widehat{\beta})$ and $\widehat{C}(\widehat{\beta})$ from Equation (13).

Tables 1–3 report the summary statistics of the simulations for the estimates of the coefficients on $X_2$, $(s/100)$ and $(s/100)^2$, respectively. We also conducted corresponding probit estimates as benchmarks. Note that, with normal errors, the probit estimates were fully efficient. The summary statistics indicated that the semiparametrically-estimated coefficients on $X_2$ were very close to the true parameter. The bias and standard deviation both decreased with sample size. This is particularly true compared to the (misspecified) probit estimator when the errors were heteroskedastic. As for the coefficient on the linear duration dependence term $(s/100)$, there appeared to be some bias, particularly in the presence of heteroskedasticity. However, the bias and RMSE of the SMS estimators diminished with sample size. This was not the case with the probit estimators. As indicated earlier, estimating duration dependence term at the $\sqrt{N}$-rate was not possible. The estimates of the coefficient on the quadratic term of the duration dependence were somewhat biased, although the bias decreased with the sample as did the RMSE. Larger sample sizes than used here may be required to estimate, with precision, more nuanced forms of duration dependence using the proposed SMS in these contexts.

**Table 1.** Simulation summary statistics—parameter: coefficient on $X_2$.

| No. of Observations | Spec (1) Normal Error | | Spec (2) Normal, Heteroscedastic Error | |
|---|---|---|---|---|
| | **500** | **1000** | **500** | **1000** |
| Using second order kernel | | | | |
| True value | 1.000 | 1.000 | 1.000 | 1.000 |
| Estimates | | | | |
|   Mean | 1.013 | 0.982 | 1.034 | 1.001 |
|   Standard dev. | 0.114 | 0.081 | 0.094 | 0.063 |
|   RMSE | 0.115 | 0.083 | 0.100 | 0.063 |
|   Skewness | 0.452 | 0.481 | 0.491 | 0.308 |
|   Kurtosis | 3.167 | 3.305 | 4.226 | 3.652 |
| Using normal cdf as continuation probability | | | | |
| True value | 1.000 | 1.000 | 1.000 | 1.000 |
| Estimates | | | | |
|   Mean | 1.017 | 1.003 | 0.937 | 0.939 |
|   Standard dev. | 0.093 | 0.032 | 0.063 | 0.045 |
|   RMSE | 0.094 | 0.032 | 0.090 | 0.076 |
|   Skewness | 0.260 | 0.114 | 0.163 | −0.082 |
|   Kurtosis | 2.712 | 2.924 | 2.900 | 2.970 |

---

[7] Estimates using a fourth-order kernel as in Horowitz (1992) yielded very similar results. The non-stochastic window-width was used, rather than, say, a plug-in window-width, to keep the simulations manageable.

**Table 2.** Simulation summary statistics—parameter: coefficient on $(s/100)$.

| No. of Observations | Spec (1) Normal Error | | Spec (2) Normal, Heteroscedastic Error | |
|---|---|---|---|---|
| | **500** | **1000** | **500** | **1000** |
| Using second order kernel | | | | |
| True value | 2.000 | 2.000 | 2.000 | 2.000 |
| Estimates | | | | |
| Mean | 2.359 | 2.112 | 1.737 | 1.790 |
| Standard dev. | 3.426 | 2.356 | 1.854 | 1.340 |
| RMSE | 3.441 | 2.356 | 1.871 | 1.355 |
| Skewness | 0.126 | 0.181 | −0.280 | −0.065 |
| Kurtosis | 4.233 | 3.986 | 8.149 | 4.617 |
| Using normal cdf as continution probability | | | | |
| True value | 2.000 | 2.000 | 2.000 | 2.000 |
| Estimates | | | | |
| Mean | 2.577 | 2.343 | 1.813 | 1.633 |
| Standard dev. | 1.544 | 1.010 | 0.830 | 0.623 |
| RMSE | 1.647 | 1.066 | 0.850 | 0.722 |
| Skewness | 0.126 | −0.042 | 0.304 | 0.433 |
| Kurtosis | 3.150 | 3.092 | 2.894 | 3.344 |

**Table 3.** Simulation summary statistics—parameter: coefficient on $(s/100)^2$.

| No. of Observations | Spec (1) Normal Error | | Spec (2) Normal, Heteroscedastic Error | |
|---|---|---|---|---|
| | **500** | **1000** | **500** | **1000** |
| Using second order kernel | | | | |
| True value | −1.000 | −1.000 | −1.000 | −1.000 |
| Estimates | | | | |
| Mean | −2.147 | −1.554 | 0.685 | 0.042 |
| Standard dev. | 14.302 | 9.805 | 6.804 | 3.911 |
| RMSE | 14.334 | 9.810 | 7.003 | 4.043 |
| Skewness | 0.182 | −3.846 | 2.400 | 1.283 |
| Kurtosis | 7.844 | 43.237 | 21.338 | 8.266 |
| Using normal cdf as continution probability | | | | |
| True value | −1.000 | −1.000 | −1.000 | −1.000 |
| Estimates | | | | |
| Mean | −4.032 | −2.678 | −1.435 | −0.922 |
| Standard dev. | 6.113 | 3.609 | 1.798 | 1.273 |
| RMSE | 6.819 | 3.977 | 1.848 | 1.274 |
| Skewness | −0.929 | −0.655 | −1.135 | −1.576 |
| Kurtosis | 4.467 | 4.051 | 5.005 | 9.012 |

We also examined the distribution of the estimates. Figures 1–3 graph the QQ-plots of the standardized SMS estimates of the coefficients on $X_2$, $s/100$ and $(s/100)^2$, respectively. Most of the standardized estimates appeared to be close to the standard normal quantiles, except for a few extreme values. The extreme values are potentially due to difficulties with numerical optimization. This would seem to indicate that the sampling distributions of the estimators in our simulation exercise were reasonably well approximated by a normal distribution.

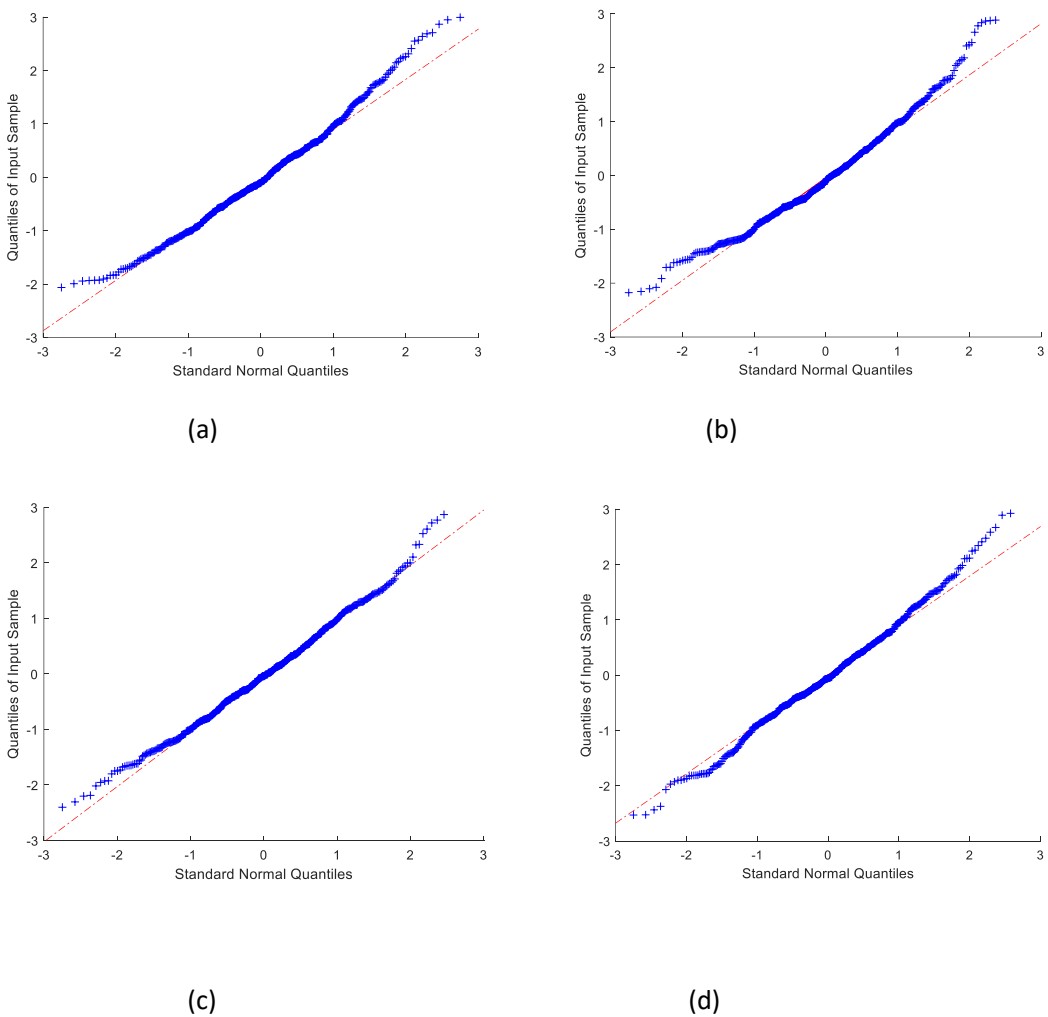

Notes: (a) Normal error, 500 observations per replication; (b) Normal error, 1000 observations per replication; (c) Heteroscedastic error, 500 observations per replication; (d) Heteroscedastic error, 1000 observations per replication.

**Figure 1.** QQ plot of estimated coefficient on $X_2$.

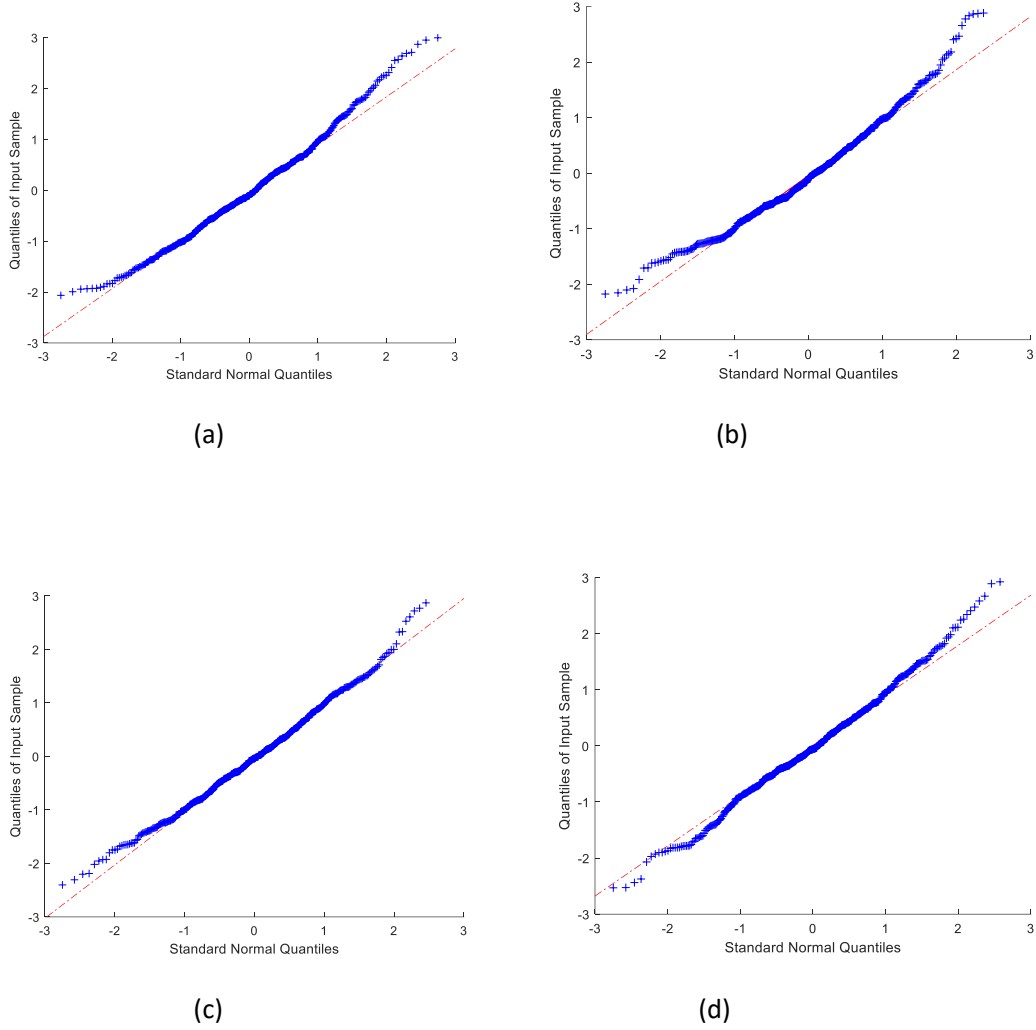

Notes: (a) Normal error, 500 observations per replication; (b) Normal error, 1000 observations per replication; (c) Heteroscedastic error, 500 observations per replication; (d) Heteroscedastic error, 1000 observations per replication.

**Figure 2.** QQ plot of estimated coefficient on $s/100$.

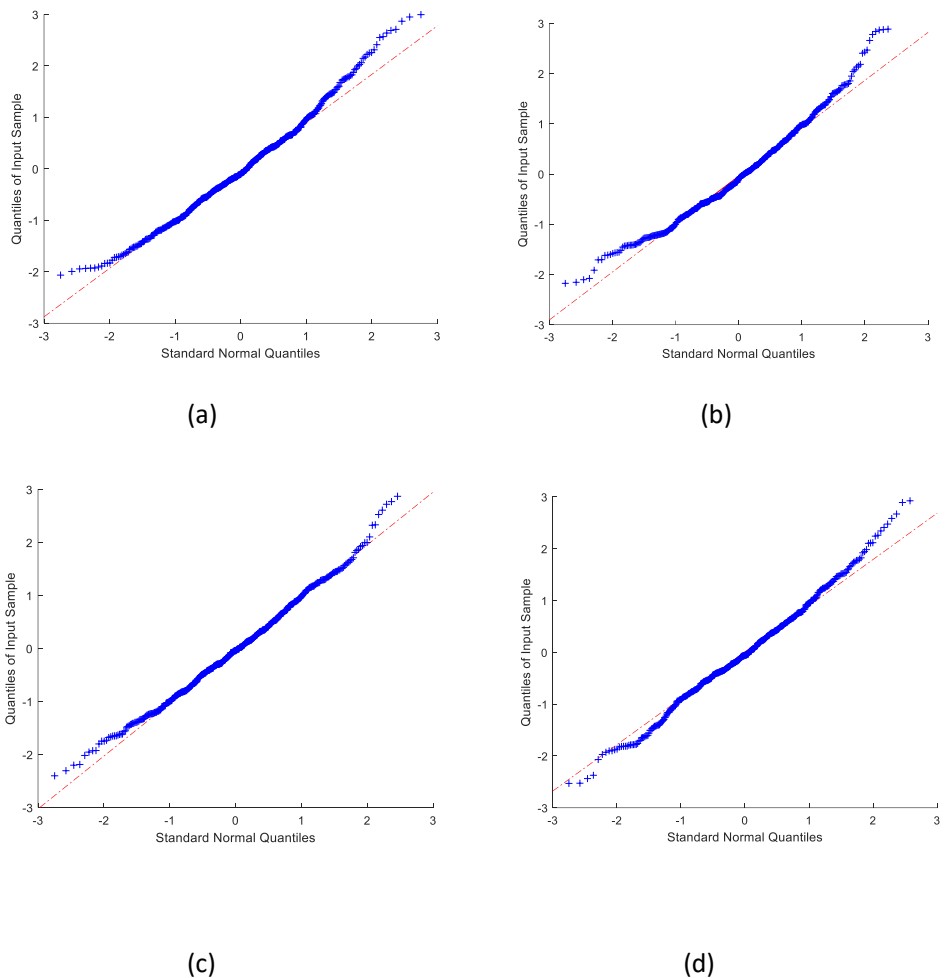

Notes: (a) Normal error, 500 observations per replication; (b) Normal error, 1000 observations per replication; (c) Heteroscedastic error, 500 observations per replication; (d) Heteroscedastic error, 1000 observations per replication.

**Figure 3.** QQ plot of estimated coefficient on $(\frac{s}{100})^2$.

## 5. Conclusions

This paper has shown that the SMS estimator can be readily adapted to consistently estimate the parameters of a popular class of discrete duration models, while relaxing the distributional assumptions of parametric models and certain semiparametric models. The asymptotic distribution of the estimators was derived and can be readily approximated using standard software. Simulations illustrated the viability of the approach. We are currently working on an empirical application of the estimator.

**Author Contributions:** Coceptualization, formal analysis, writing, P.R.; software, validation, S.R.

**Funding:** This research received no external funding.

**Acknowledgments:** The authors appreciate comments from Richard Blundell, Christian Bontemps, Juan Rodríguez-Poo and seminar participants at the 2016 African Meetings of the Econometric Society in South Africa and the 2017 International Conference on Panel Data in Thessaloníki. The authors are responsible for any errors.

**Conflicts of Interest:** The authors declare no conflict of interest.

## Appendix A

**Proof of Lemma 1 (a).** To derive the expected value of $q_{is}^{(1)}(\beta_0)$, suppress the $i$ subscripts, and write:

$$
\begin{aligned}
\mathbb{E}[q_s^{(1)}|X_s, Y_{s-1}] &= \mathbb{E}[Y_{s-1}(2Y_s-1)K^{(1)}(Z_s/\gamma)X_s X_s^\top]/\gamma^2 \\
&= \mathbb{E}[A_s|X_s, Y_{s-1}]X_s X_s^\top Y_{s-1}
\end{aligned}
\tag{A1}
$$

where $A_s = (21[Z_s + U_s \geq 0] - 1)\frac{1}{\gamma^2}K^{(1)}(Z_s/\gamma)$, suppressing the $X_s$ and $Y_{s-1}$ arguments in $g(u_s|z_s, x_s, y_{s-1})$ and $h(z_s|x_s, y_{s-1})$.

$$
\begin{aligned}
\mathbb{E}[A_s|Z_s] &= \int (21[Z_s + u_s \geq 0] - 1)K^{(1)}(Z/\gamma)g(u_s|Z_s)du_s/\gamma^2 \\
&= \left(\int_{-Z_s}^{\infty} + \int_{-\infty}^{-Z_s}\right)(21[Z_s + u_s \geq 0] - 1)K^{(1)}(Z_s/\gamma)g(u_s|Z_s)du_s/\gamma^2 \\
&= K^{(1)}(Z_s/\gamma)\left(\int_{-Z_s}^{\infty} - \int_{-\infty}^{-Z_s}\right)g(u_s|Z_s)du_s.
\end{aligned}
\tag{A2}
$$

$$
\begin{aligned}
\mathbb{E}[A_s] &= \int K^{(1)}(z_s/\gamma)\left(\int_{-z_s}^{\infty} - \int_{-\infty}^{-z_s}\right)g(u_s|z_s)du_s/\gamma^2 f(z_s)dz_s \\
&= \int K^{(1)}(w)\left(\int_{-w\gamma}^{\infty} - \int_{-\infty}^{-w\gamma}\right)g(u_s|w\gamma)f(w\gamma)du_s dw/\gamma \\
&= \int K^{(1)}(w)\left((1 - G(-w\gamma|w\gamma) - G(-w\gamma|w\gamma)\right)f(w\gamma)dw/\gamma \\
&= \int K^{(1)}(w)\left((1 - 2G(-w\gamma|w\gamma)\right)f(w\gamma)dw/\gamma \\
&= -\int K(w)(((1 - 2G(-w\gamma|w\gamma))f(w\gamma))^{[1]}dw \\
&\to -(((1 - 2G(-z_s|z_s))f(z_s))^{[1]}_{z_s=0} \\
&= 2G^{[1]}(0|0)f(0)
\end{aligned}
\tag{A3}
$$

To prove Part (b), make substitutions as in (a), with:

$$
\mathbb{E}[q_s|X_s, Y_{s-1}] = \mathbb{E}[A_s|X_s, Y_{s-1}]X_s Y_{s-1}
\tag{A4}
$$

where $A_s = (21[Z_s + U_s \geq 0] - 1)K(Z_s/\gamma)/\gamma$.

$$
\begin{aligned}
\mathbb{E}[A_s|Z_s] &= \int (21[Z_s + u_s \geq 0] - 1)K(Z_s/\gamma)g(u_s|Z_s)du_s/\gamma \\
&= K(Z_s/\gamma)\left(\int_{-Z_s}^{\infty} + \int_{-\infty}^{-Z_s}\right)(21[Z_s + u_s \geq 0] - 1)g(u_s|Z_s)du_s/\gamma \\
&= K(Z_s/\gamma)\left(\int_{-Z_s}^{\infty} - \int_{-\infty}^{-Z_s}\right)g(u_s|Z_s)du_s/\gamma \\
&= K(Z_s/\gamma)\left(1 - 2G(-Z_s|Z_s)\right)/\gamma
\end{aligned}
\tag{A5}
$$

so that:

$$
\begin{aligned}
\mathbb{E}[A_s] &= \int K(z_s/\gamma)\left(1 - 2G(-z_s|z_s)\right)f(z_s)dz_s/\gamma \\
&= \int K(w)\bar{A}(w\gamma)dw
\end{aligned}
$$

where $\bar{A}(\gamma) = (1 - 2G(-w\gamma|w\gamma))f(w\gamma)$ and:

$$\int K(w)\bar{A}(w\gamma)dw = \int K(w)\left(\bar{A}(0) + \sum_{j=1}^{s-1}\frac{\bar{A}^{[j]}(0)(w\gamma)^j}{j!} + \frac{\bar{A}^{[m]}(\bar{\gamma})(w\gamma)^m}{m!}\right)dw. \tag{A6}$$

Note that $\bar{A}(0) = 0$ and all the middle terms in Equation (A6) are zero from $\int w^j K(w)dw = 0$, $j = 1, \ldots, m - 1$. As for the third term, first note that:

$$\int K(w)\left(\bar{A}^{[m]}(\bar{\gamma}) - \bar{A}^{[m]}(0)\right)w^m dw\bar{A}^{[m]}(\gamma) = o(1) \tag{A7}$$

by dominated convergence, uniformly on $x_s, Y_{s-1}$. There are a few ways to write $\bar{A}^{[m]}(0)$. It is simplest to note first that:

$$\int K(w)A^{[m]}(0)w^j dw = \mu_m A^{[m]}(0) \tag{A8}$$

and by the binomial theorem:

$$\begin{aligned}
A^{[m]}(0) &= \sum_{j=1}^{m}\binom{m}{j}(1 - 2G(-u|u))^{[m-j]}f^{[j]}(z)\Big|_{u=0} \\
&= -2\sum_{j=1}^{m-1}\binom{m}{j}G^{[m-j]}(-z|z)f^{[j]}(u)\Big|_{z=0}.
\end{aligned} \tag{A9}$$

To prove Part (c):

$$\gamma\mathbb{E}[q_s q_\tau^\top | X_s, Y_{s-1}, X_\tau, Y_{\tau-1}] = \mathbb{E}[A_{s\tau}|X_s, Y_{s-1}, X_\tau, Y_{\tau-1}]X_s Y_{s-1}X_\tau^\top Y_{\tau-1} \tag{A10}$$

where $A_{s\tau} = (21[Z_s + U_s \geq 0] - 1)K(Z_s/\gamma)(21[Z_\tau + U_\tau \geq 0] - 1)K(Z_\tau/\gamma)/\gamma^2$. From Assumption 1, we have:

$$\mathbb{E}[A_{s\tau}|X_s, Y_{s-1}, X_\tau, Y_{\tau-1}] = \begin{cases} \mathbb{E}[(21[Z_s + U_s \geq 0] - 1)^2 K(Z_s/\gamma)^2)/\gamma^2 | X_s, Y_{s-1}], & s = \tau \\ (\mathbb{E}[(21[Z_s + U_s \geq 0] - 1)^2 K(Z_s/\gamma)/\gamma | X_s, Y_{s-1}])^2 = O(1), & s \neq \tau. \end{cases} \tag{A11}$$

It suffices to only consider when $s = \tau$, as it converges at a slower rate than when $s \neq \tau$.

$$\begin{aligned}
\mathbb{E}[A_{s\tau}|Z_s] &= K(Z_s/\gamma)^2 \int (21[Z_s + u_s \geq 0] - 1)^2 g(u_s|Z_s)du_s/\gamma^2 \\
&= K(Z_s/\gamma)^2 \left(\int_{-\infty}^{-Z_s} + \int_{-Z_s}^{\infty}\right)(21[Z_s + u_s \geq 0] - 1)^2 g(u_s|Z_s)du_s/\gamma^2 \\
&= K(Z_s/\gamma)^2 \left(\int_{-\infty}^{-Z_s} + \int_{-Z_s}^{\infty}\right)g(u_s|Z_s)du_s/\gamma^2
\end{aligned} \tag{A12}$$

so that:

$$\begin{aligned}
\mathbb{E}[A_{s\tau}] &= \int K(z_s/\gamma)^2 \left(\int_{-\infty}^{-z_s} + \int_{-z_s}^{\infty}\right)g(u_s|z_s)du_s/\gamma^2 f(z_s)dz_s/\gamma^2 \\
&= \int K(z_s/\gamma)^2 \left((1 - G(u_s|s_s) + G(u_s|z_s))\right)f(z_s)dz_s/\gamma^2 \\
&= \int K(w)^2 f(w\gamma)dw/\gamma
\end{aligned}$$

and $\gamma\mathbb{E}[A_{s\tau}] \to f(0)\int K(w)^2 dw$. $\quad\square$

**Lemma A1.** *Assume $\bar{\beta} \xrightarrow{p} \beta_0$. Then, under Assumptions 1–11, $\psi_N^{(1)}(\bar{\beta}) = Q + o_p(1)$.*

**Proof of Lemma A1.** For $\psi_N^{(1)}(\bar{\beta})$, note that by the uniform law of large numbers and Slutsky's theorem, $\psi_N^{(1)}(\bar{\beta}) \to \lim_{N\to\infty} \mathbb{E}[q_i^{(1)}(\beta_0)] = Q.$  □

**Proof of Proposition 1.** (a) Consistency is shown by combining and extending the results of Manski (1985) and Horowitz (1992). Following Manski, define a population objective function $\Psi^*(\beta) = \sum_{s-1}^{S}(2\Pr(Y_s = 1, Z_s(\beta) \geq 0 | Y_{s-1}) - \Pr(Z_s(\beta) \geq 0 | Y_{s-1}))\Pr(Y_{s-1} = 1).$ [8] As per Manski, $\Psi^*(\beta)$ is maximized uniquely at $\beta = \beta_0$, is continuous and $\Psi_N^*(\beta)$ converges uniformly to $\Psi^*(\beta)$. Extending Horowitz, we have $|\Psi_N^*(\beta) - \Psi_N(\beta)| \xrightarrow{p} 0$ uniformly in $\beta$, and hence, $\widehat{\beta}$ is consistent. (b) To derive the asymptotic distribution, use a Taylor series expansion of the first-order conditions, rearranging them so that:

$$\sqrt{N\gamma}(\widehat{\beta} - \beta_0) = (\psi_N^{(1)}(\bar{\beta}))^{-1}\sqrt{N\gamma}(\psi_N(\beta_0) - \mathbb{E}\psi_N(\beta_0)) \tag{A13}$$

and from Lemmas 1 and A1:

$$\sqrt{N\gamma}(\widehat{\beta} - \beta_0 - \gamma^m Q^{-1}B) = (Q^{-1} + o_P(1))\sqrt{N\gamma}\frac{1}{N}\sum \tilde{q}_i + o_P(\sqrt{N\gamma}\gamma^m). \tag{A14}$$

Application of the central limit theorem completes the result.    □

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
