# Peer review of "Smoothed Maximum Score Estimation of Discrete Duration Models"

_jrfm, doi:10.3390/jrfm12020064_

Round 1

Reviewer 1 Report

This paper considers the estimation of discrete-time duration models. It proposes a new method using a (extension of) Smoothed Maximum Score Estimator. Asymptotic properties are derived and a Monte Carlo study of the finite properties is performed.

The math seems correct and so does the proofs but the paper is still missing something: I read it and don’t know why I should care!

·        The very first line of the introduction says that there are many ways to estimate this type of models. So why should I care about this one? The results, in particular the finite sample ones, are never benchmarked against any existing methods.

·        More generally, what is I can do with this type of estimator that I could not “do” before and why is this of any (economic) importance?

·        Even for a special issue on nonparametric econometric method I believe some background and context is needed to show the importance of this.

·        And there is not really any application – again I fail to see this at least.

·        The paper cites Reza and Rilstone (2013 and 2015) yet neither paper appear in the reference list.

·        Correction on page 8 line 239 is misspelled.

Author Response

Referee 1 has raised some very useful points. We have worked hard to respond to these   as follows.

We have substantially rewritten the Introduction to explain more clearly in which way our paper makes a contribution. We contextualize discrete duration models as extensions of single-period binary choice models. Manski and Horowitz type estimators which relax moment and smoothness conditions inherent in the Ichimura and Klein-Spady approaches do not as yet have a discrete duration extension.  The referee commented on the absence of an empirical application. Since we conducted a simulation study to examine the practical feasibility of the estimator, we did not consider this necessary. We are currently working on an empirical application and we make a note of  this now in the paper's conclusion. As per the referee's comment, we also   estimate parametric benchmark (probit, which are efficient with the homoskedastc errors) and  compare them to the proposed estimator. The proposed SMS estimator does comparitively  well   estimating the coefficients on the covariate X and s, a little less well with respect to s2.The references  to Reza and Rilstone (2014, 2016) are now provided. The typo has been corrected.

Reviewer 2 Report

Please read the file attached.

Author Response

Referee 2 made numerous useful remarks.  x*   should  appear as a conditioning variable in Assumption 3 and has been added.  x*  does not appear as a conditioning variable in other locations when  we state conditions re  z  where  z = x* s + x'b. In those cases the conditional distribution of z would be degenerate.  The Reza and Rilstone reference is included now. The maximum observed duration across all the simulations was 37. Implicitly we set  S=37. We note this in the revision. Choice of  S is not an issue as a practical matter since one can simply write the log-likelihood to sum to the final observed period for each individual.

Reviewer 3 Report

see the attachment

Author Response

Referee 3 made a number of  useful remarks. We have included a note to explain why we choose to introduce the identification restriction on the coefficient on the first covariate when defining the class of models under consideration. The two typos have been fixed. While for a given empirical application it would make sense to use a plug-in kernel, we did not find this viable during the simulations. Our simulations required optimization in several directions, requiring a very large amount of time to begin with. We have included a note in this regard.

Round 2

Reviewer 1 Report

NA